# The role of noninfectious comorbidities in the association between severe infections and risk of dementia in Finland: A nationwide registry study

Pyry N. Sipilä[1*], Kaarina Korhonen[2,3], Joni V. Lindbohm[1,4,5], Mika Kivimäki[1,4‡], Pekka Martikainen[2,3,6‡]

1 Department of Public Health, Clinicum, University of Helsinki, Helsinki, Finland, 2 Helsinki Institute for Demography and Population Health, University of Helsinki, Helsinki, Finland, 3 Max Planck-University of Helsinki Center for Social Inequalities in Population Health, University of Helsinki, Helsinki, Finland, 4 Brain Sciences, University College London, London, United Kingdom, 5 The Klarman Cell Observatory, Broad Institute of MIT and Harvard, Cambridge, Massachusetts, United States of America, 6 Laboratory of Population Health, Max Planck Institute for Demographic Research, Rostock, Germany

‡ These authors are joint senior authors on this work.
* pyry.sipila@helsinki.fi

## Abstract

### Background

Severe infections have been linked to an increased risk of dementia, but both conditions often coexist with other illnesses that may confound this association. Using nationwide Finnish health registry data, we examined the role of noninfectious mental and physical illnesses in the association between severe infections and dementia.

### Methods and findings

This register-based study included 62,555 individuals aged 65 or older in Finland in 2016 who were diagnosed with late-onset dementia between 2017 and 2020 and 312,772 dementia-free controls matched for year of birth, sex, and the follow-up period. Analyses were adjusted for education, marital status, employment, and area of residence, with age and sex accounted for through the matched conditional design and analysis. Applying a 1-year lag period, we identified 29 hospital-treated diseases that occurred 1–21 years before dementia diagnosis in cases (or index date in controls), had a prevalence of ≥ 1% prior to dementia, and were robustly associated with increased dementia risk (confounder-adjusted rate ratio ≥ 1.20, $p < 0.000294$). In addition to 2 infectious diseases (cystitis and bacterial infection of an unspecified site), these included 27 mental, behavioural, digestive, endocrine, cardiometabolic, neurological, and eye diseases, as well as injuries. 29,376 (47%) of the dementia cases had at least one of these diseases diagnosed before dementia. The associations between the two infectious diseases and dementia risk were

**Data availability statement:** Data sharing: Due to data protection regulations of the national register-holders providing the data, we are not allowed to make the data available to third parties. Interested researchers have the possibility to obtain data access by contacting the following register-holding public institutions: Statistics Finland (https://stat.fi/tup/tutkijapalvelut/index_en.html; tutkijapalvelut@stat.fi) and the Finnish Social and Health Data Permit Authority Findata (https://findata.fi/en/permits/; info@findata.fi). Statistical codes used in analysis are provided in S1 Stata Codes.

**Funding:** PNS was supported by the Emil Aaltonen Foundation (https://emilaaltonen.fi/apurahat/in-english/), the Finnish Medical Foundation (https://laaketieteensaatio.fi/en/home/), and the Päivikki and Sakari Sohlberg Foundation (https://pss-saatio.fi/en/) during the conduct of the study. J.V.L. was supported by the Research Council of Finland (339568, https://www.aka.fi/en/) and the Päivikki and Sakari Sohlberg Foundation. MK was supported by the Wellcome Trust (221854/Z/20/Z, https://wellcome.org/), the UK Medical Research Council (MR/Y014154/1, https://www.ukri.org/councils/mrc/), the National Institute on Aging (National Institutes of Health), USA (R01AG056477, R01AG062553; https://www.nia.nih.gov/) and the Research Council of Finland (350426). PM was supported by the European Research Council under the European Union's Horizon 2020 research and innovation programme (grant agreement No 101019329, https://erc.europa.eu/homepage), the Strategic Research Council (SRC) within the Research Council of Finland grants for ACElife (#352543-352572) and LIFECON (# 345219), the Research Council of Finland profiling grant for SWAN and FooDrug, and grants to the Max Planck – University of Helsinki Center from the Jane and Aatos Erkko Foundation (#210046, https://jaes.fi/en/frontpage/), the Max Planck Society (# 5714240218, https://www.mpg.de/en), University of Helsinki (#77204227, https://www.helsinki.fi/en), and Cities of Helsinki (https://www.hel.fi/en), Vantaa (https://www.vantaa.fi/en) and Espoo (https://www.espoo.fi/en). Open access funded by Helsinki University Library to PNS (https://www.helsinki.fi/en/helsinki-university-library). The funders had no role in the study design, data collection and

not attributable to the 27 comorbid dementia-related diseases diagnosed before infections. The adjusted rate ratio for cystitis was 1.22 (95% confidence interval (CI) [1.17, 1.27]; $p < 0.001$) before and 1.19 (95% CI [1.14, 1.24]; $p < 0.001$) after adjustment for comorbidities, while for bacterial infections of an unspecified site, the rate ratios were 1.21 (95% CI [1.16, 1.28]; $p < 0.001$) and 1.19 (95% CI [1.13, 1.25]; $p < 0.001$), respectively. The findings were comparable across subgroups defined by sex and education, and stronger for cases of early onset dementia. We were not able to directly assess psychosocial, behavioural, or biological confounders that are not captured in nationwide registries.

## Conclusions

This nationwide Finnish study identified several mental and physical diseases that are associated with an increased risk of dementia and showed that the increased incidence of dementia among individuals with severe infections is not attributable to these comorbid conditions. These results support the role of severe infections as independent risk factors for dementia.

---

## Author summary
### Why was this study done?

- People who experience severe infections are at higher risk of developing dementia.

- It is unclear whether this association is explained by other coexisting (noninfectious) diseases that predispose to both infections and dementia.

- A better understanding of dementia risk factors is needed to inform more effective strategies for dementia prevention.

### What did the researchers do and find?

- Using nationwide Finnish registers, we identified 62,555 individuals aged 65 or older who were diagnosed with late-onset dementia between 2017 and 2020, and 312,772 dementia-free controls matched for year of birth, sex, and the follow-up period.

- We found that in a period of 1–21 years before follow-up, 29 hospital-treated diseases were associated with an increased risk of developing dementia, many of which were also interrelated.

- Two of the 29 diseases were infections, and both remained independently associated with dementia risk after adjustment for all other dementia-related diseases.

analysis, decision to publish, or preparation of the manuscript.

**Competing interests:** The authors have declared that no competing interests exist.

**Abbreviations:** ATC, Anatomical Therapeutic Chemical; ICD-9, International Classification of Diseases, 9th revision; CI, confidence interval; ERE, excess risk explained; ICD-10, International Classification of Diseases, 10th revision; ISCED, International Standard Classification of Education; RECORD, REporting of studies Conducted using Observational Routinely-collected health Data; RR, rate ratio; SD, standard deviation.

## What do these findings mean?

- Many interconnected diseases are associated with an increased risk of dementia.

- Severe infections, although linked to several other dementia-related diseases, are independently associated with higher dementia risk.

- Because this study is observational, intervention trials are required to establish whether better infection prevention could reduce dementia incidence.

## Introduction

Growing evidence from observational cohort studies suggests that severe (hospital-treated) infections are associated with an increased risk of dementia [1–6]. Several plausible mechanisms have been proposed to explain this association, including disturbed peripheral-central immune crosstalk, which contributes to neuroinflammation [7,8]; inflammation-induced blood-brain barrier dysfunction and related entry of neurotoxic plasma components, blood cells and pathogens into the central nervous system [9]; and infection-related vascular mechanisms, such as platelet activation, inflammation-induced thrombosis, and endothelial dysfunction in the brain [10,11].

However, given its typical late-onset, clinically diagnosed dementia rarely occurs in isolation from other age-related diseases [12,13]. Longitudinal studies have identified cascades of physical and mental diseases that develop over several years before clinical dementia manifests [14,15]. Many of these diseases also increase the risk of infections [16–20]. To date, few studies have examined the extent to which comorbid conditions, such as diabetes, cardio- and cerebrovascular diseases, and depression, contribute to the excess dementia risk observed among individuals with severe infections. A major barrier for this research is the need for large population samples and extended follow-ups to track disease progression over time. As a result, it remains unclear whether severe infections independently increase the risk of dementia.

We addressed this knowledge gap in the following ways. First, to ensure comprehensive analysis, we applied a phenome-wide approach, including all hospital presenting diseases or disease groups as defined in the International Classification of Diseases, 10th revision (ICD-10), with a prevalence of 1% or higher among dementia cases. We assessed which of these diseases were associated with an increased risk of dementia among dementia cases and controls drawn from the total population of Finland. Second, to examine the sequence, timing, and inter-relatedness of the identified dementia-related diseases, we analysed disease trajectories and disease networks over 20 years preceding dementia onset, focussing on conditions that preceded both severe infections and dementia. Third, we assessed the extent to which the association between severe infections and dementia risk remains after accounting for other dementia-related diseases. We anticipated identifying several infectious

and noninfectious diseases that are interrelated and associated with dementia risk in Finnish adults. We hypothesised that severe infections increase the likelihood of developing dementia at least partly independently of preceding noninfectious comorbidities.

## Methods

### Study population and register linkage

This study focussed on the Finnish population aged 65+ (born 1951 or earlier), free of dementia, and residing in Finland on 31 December 2016. A supplementary analysis of early onset dementia was conducted among the total Finnish population aged 18–64 during the study period (born 1953–1977). Statistics Finland provided demographic and mortality data. Using unique personal identification codes, Statistics Finland linked the study individuals to hospitalisation records from the Care Register of the Finnish Institute for Health and Welfare and to medication reimbursement and purchase data recorded by the Social Insurance Institution of Finland.

All the data were pseudonymised before release to researchers and data analysis was approved by Statistics Finland Board of Ethics (permit no. TK/1836/07.03.00/2024) and the Social and Health Data Permit Authority Findata (permit no. THL/6846/14.02.00/2024). Informed consent was not required for this study, because only administrative register data were used (the Finnish Personal Data Act and the Statistics Act).

### Study design and sampling of controls

This observational study includes individuals aged 65 or older living in Finland in 2016 who were diagnosed with late-onset dementia between 2017 and 2020 and who also resided in Finland at the start of the exposure period on 1 January 1996–1 January 1999 (residence recorded on 31 December 1995–31 December 1998). Immigrants who arrived after the baseline of 1996–1999 were excluded. Using incidence-density sampling with replacement, we aimed to identify 5 controls for each dementia case, matching exactly for year of birth, sex, and follow-up period. Controls were required to be at risk (i.e., dementia-free) at the time of the matched case's dementia diagnosis (the index date) but could later become cases if diagnosed with dementia after that date. Five suitable controls were found for almost all cases, resulting in a final sample of 62,555 dementia cases and 312,772 controls. Those who died were censored from the risk pool on the last day of the year of death. Emigration after 2016 was very rare (~0.05% annually) [21]. Therefore, we assumed that all study individuals at the end of 2016 continued living in Finland until the end of 2020 unless they died. The incidence-density sampling approach guarantees that controls are sampled from person-time at risk and, therefore, the resulting odds ratios from matched analysis represent rate ratios (RR) [22,23]. In additional analyses, we also examined early onset dementia (age of onset <65 years) using a similar sampling scheme.

### Hospitalisations from severe infections or other nondementia illnesses (exposures)

Using hospital inpatient and outpatient records, we ascertained all diseases recorded as a primary diagnosis of the encounter during a two-decade exposure window 1–21 years before dementia diagnosis for dementia cases and the index date for matched controls (Fig 1). The exposure window was set at 21 years, corresponding the longest period for which ICD-10 data were available. A long exposure period was necessary because dementia typically develops over many years or even decades [24]. Diagnoses recorded <1 year before dementia onset or index date were not included to reduce bias from delays in the recording of the dementia diagnoses. Thus, depending on the year of dementia diagnosis or index date in 2017–2020, the timespan for disease ascertainment started between 1 January 1996 and 1 January 1999 and ended between 31 December 2015 and 31 December 2018, respectively. We analysed each 3-digit ICD-10 disease code as a separate entity and did not attempt to group codes that may refer to overlapping or clinically similar conditions, including ICD-10 chapters I–XIV, XIX, and XX, which contain diseases, injuries, and related conditions.

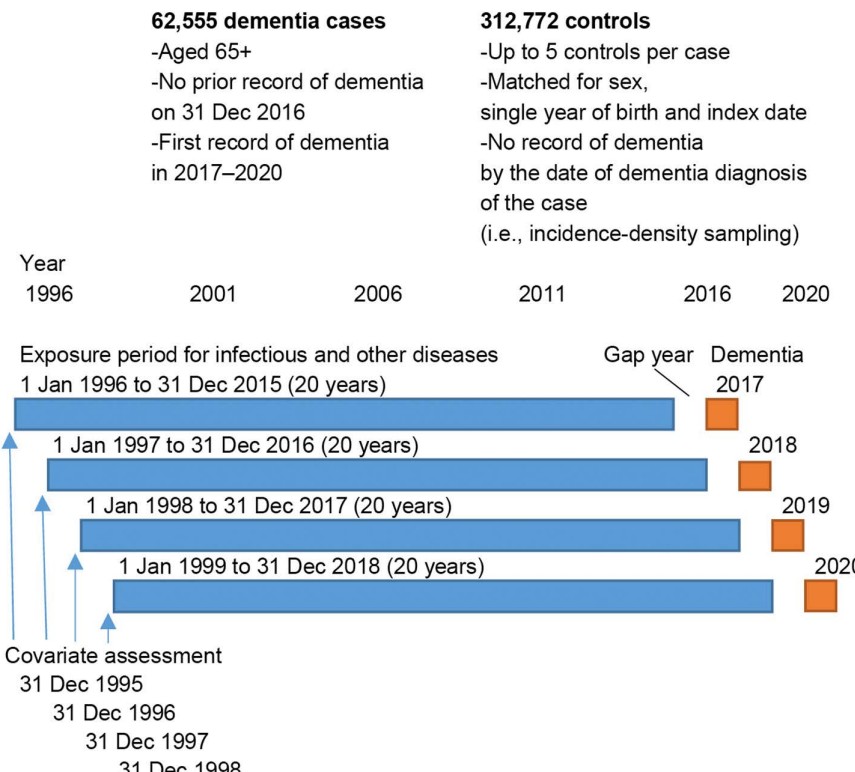

**Fig 1. Study design.** The figure outlines the criteria used to select dementia cases and matched controls and illustrates the study design, including baseline covariate assessment, the 20-year exposure window for identifying infectious and noninfectious dementia-related diseases, the gap year before dementia diagnosis, and the dementia diagnosis year (2017–2020).

Chapters XV, XVI, XVII, XVIII, XXI, and XXII were not included, because they mostly refer to nondisease events, such as pregnancy and childbirth, perinatal period, congenital malformations, unspecific symptoms, administration, and other special purposes.

### Ascertainment of all-cause dementia cases

We retrieved diagnoses of dementia from three sources: primary and secondary diagnoses in hospital inpatient and outpatient records; medication reimbursement entitlements for the treatment of dementia; and purchases of prescribed antidementia medications (the Anatomical Therapeutic Chemical [ATC] code N06D). We defined all-cause dementia using the ICD-10 codes F00–F03, F05.1, G30, G31.0, G31.1, and G31.8 [1]. Incident cases of dementia were recorded between 1 January 2017 and 31 December 2020. Individuals with a record of prevalent all-cause dementia between 1 January 1996 (the inception of ICD-10 in Finland) and 31 December 2016 were excluded. To further ensure that the individuals did not have prevalent dementia at the start of the exposure window (from 1 January 1996 to 1 January 1999), we also excluded individuals who had a record of any dementia-related 9th revision (ICD-9) codes (290, 2900A, 2941A, 3310A, 3311A, 3312X, 3318X, and 4378A) between 1 January 1987 and 31 December 1995, the time-period when ICD-9 was in use in Finland.

Hospital inpatient records were included from the beginning of the use of ICD-9 in Finland in 1987 and were available until 2020. Hospital outpatient records were available from 1998 to 2020, reimbursement data for antidementia medications from 1999 to 2018, and prescribed medication purchase data for antidementia medications from 2000 to 2019.

## Covariates

We considered sex, education, marital status, employment, and area of residence as covariates. These data were obtained from Statistics Finland registries. We used the first records immediately before the start of disease ascertainment in 1996–1999. Based on International Standard Classification of Education (ISCED) 2011, highest achieved education was categorised into three groups for adjustments (tertiary; secondary; basic or no qualifications) and dichotomised into high versus low for stratified analyses (tertiary or secondary versus basic or no qualifications). Marital status was recorded as married, never married, divorced, and widowed. Employment status was recorded as employed (aged<60); retired early (aged<60); other nonworking (aged<60); entitled to old-age retirement (aged 60+) [25]. In Finland, public healthcare covers a great majority of hospitalisations and is organised on a regional basis. To account for potential regional differences in healthcare, we adjusted our models for area of residence using 19 dummy variables which corresponded to the administrative regional division confirmed by the Finnish Government. The three largest regions by population were Uusimaa (comprising the capital Helsinki and surrounding areas), Pirkanmaa (Tampere and surrounding areas), and Southwest Finland (Turku and surrounding areas). In sensitivity analyses, we additionally examined household income quintiles as defined by Statistics Finland and use of medications for hypertension, diabetes, and coronary heart disease based on medication reimbursement entitlements from the Social Insurance Institution of Finland as covariates.

## Statistical analysis

In the first part of the data analysis, we identified diseases that were associated with an increased risk of incident dementia. This was done through three steps. First, we identified all diseases that preceded dementia with a prevalence of at least 1% during the exposure window (1–21 years before dementia diagnosis). As in genetic studies of common versus rare variants, we chose the 1% threshold to focus on relatively common diseases that have relevance on the population health level [26]. Second, we analysed the associations between all these diseases and incident dementia using conditional logistic regression, adjusting for education, marital status, employment, and area of residence. Adjustment for age and sex was inherent to the matched conditional design and analysis, as matching was performed on year of birth, index date, and sex [27]. Third, diseases showing a statistically significant association with dementia after Bonferroni correction ($p<0.000294$) and an RR of at least 1.20 were selected for further analysis. The RR threshold of 1.20 was chosen based on literature indicating that smaller effect sizes are often of limited public health relevance and more susceptible to bias, unless the disease incidence is very high [15,28,29].

In the second part of the data analysis, we constructed disease trajectories and networks across dementia-related diseases in four steps. First, we identified associations between all the possible pairs of dementia-related diseases identified in the first part of the data analysis. This was done among the dementia cases using the same 20-year exposure window as with the dementia analysis (from 1 January 1996–1 January 1999 to 31 December 2015–31 December 2018). Because any pair of diseases can occur in two temporal sequences (A→B or B→A), we first determined which sequence occurred more often in the data. That sequence was used in all primary trajectory analyses, while the alternative sequence was assessed in supplementary analyses. Second, as in the analysis for incident dementia, we used incidence-density sampling for each disease, identifying up to 5 controls per disease case, matched for year of birth, index date, and sex. For virtually all disease cases, five appropriate controls were identified, with incomplete matches only occurring in ≤0.4% of cases. Third, after establishing the analytic design in the previous steps, we examined the strength of the associations between disease pairs using conditional logistic regression adjusted for education, marital status, employment, and area of residence, and for age and sex inherently to the matched conditional design and analysis. *P*-values were adjusted for multiple testing using Bonferroni correction. Fourth, Bonferroni significant disease pairs in which the first disease had an RR of ≥1.20 for the occurrence of the second disease were retained in analysis. From these dementia-related disease pairs, we constructed disease trajectories in which diseases were represented as nodes and their interconnections as arrows preserving the predominant temporal sequence [15]. For example, an arrow from depression to alcohol use

disorder indicates that depression diagnosis more commonly preceded the diagnosis of alcohol use disorder in the data and that a diagnosis of depression was associated with an increased subsequent risk of alcohol use disorder.

In the third part of the data analysis, we repeated the analyses on the associations between infections and incident dementia, additionally adjusting for any other dementia-related diseases. To minimise the risk of residual confounding, these additional adjustments included all noninfectious dementia-related diseases regardless of whether they were components of the disease trajectories leading to infections. We also assessed the associations of infections with dementia in subgroups defined by sex and education (tertiary or secondary [high] versus basic or no qualifications [low]). We conducted several sensitivity analyses to test the robustness of our findings; we 1) included additional adjustments for income and prior medication use, 2) excluded all controls who developed dementia after the index date, 3) used a year of birth, index year, and sex-adjusted mixed effects logistic regression model with random effects for area of residence instead of the conditional logistic regression used in the main analyses, 4) and used inverse-probability weighting to account for potential confounding by all the variables used in the most adjusted model (year of birth; index year; sex; education; marital status; employment; area of residence; income; medications for hypertension, diabetes, and coronary heart disease; noninfectious dementia-related diseases). We also conducted 5-year lag-analyses in which we ignored infections and other diseases that were diagnosed less than 5 years before dementia (or index date for controls).

From the adjusted analyses, we computed excess risk explained by comorbidities (ERE) as

$$ERE = \frac{RR\_adjusted\_for\_covariates - RR\_adjusted\_for\_covariates\_and\_comorbidities}{RR\_adjusted\_for\_covariates - 1}$$

Data were analysed using Stata MP 17 and 18. Statistical codes used in analysis are provided in S1 Stata Codes. Confidence intervals (CI) are reported at the 95% level. The analyses were based on a pre-specified analysis plan (S1 Study Plan). Amendments were later made to increase the comprehensiveness of disease identification (RR threshold changed from ≥ 1.50 to ≥ 1.20), to add analyses for early onset dementia, to add adjustments, to combine high and intermediate education in stratified analyses to gain power, and to add sensitivity analyses in response to peer review comments. This study is reported in accordance with the REporting of studies Conducted using Observational Routinely-collected health Data (RECORD) guideline (S1 RECORD Checklist).

## Results

We identified 62,555 dementia cases and 312,772 control individuals matched for year of birth, sex, and follow-up period. The mean age was 81.0 (standard deviation [SD] 7.0) among both the cases and controls. 37,582 (60.1%) of the dementia cases were women, 37,042 (59.2%) had at most basic education, and 41,904 (67.0%) were married (Table 1). The corresponding numbers were 187,910 (60.1%), 177,825 (56.9%), and 217,375 (69.5%) among the controls.

In the first part of the analysis, we identified diseases robustly associated with incident dementia. Of the 170 diseases that preceded dementia with a prevalence of at least 1% during the 1–21-year exposure window, 59 showed statistically significant associations after Bonferroni correction for 170 tests ($p < 0.000294$, Fig 2). Of these, 29 had an RR of at least 1.20 for dementia (Fig 3). These included 9 injuries, 6 mental and behavioural disorders, 5 cardiovascular and 3 neurological diseases, 1 endocrine, 1 metabolic, 1 eye and 1 digestive disease, as well as 2 infections (cystitis [an infection of the genitourinary system] and bacterial infection of unspecified site). Overall, 29,376 (47.0%) of the 62,555 dementia cases had at least one of these diseases during the two-decade exposure period before dementia, and 12,896 (20.6%) had at least two (Table A in S1 Appendix). The most common were cerebral infarction (prevalence 9.6%), intracranial injury (6.0%), and type 2 diabetes mellitus (4.9%). Numbers of dementia cases and controls with infections by type of infection are shown in Table 2.

The strongest associations with dementia were seen for mental disorders due to brain damage or physical disease (RR 3.76, 95% CI [3.48, 4.06]; $p < 0.001$), Parkinson's disease (RR 3.24, 95% CI [3.06, 3.44]; $p < 0.001$), and alcohol-related mental and behavioural disorders (RR 1.87, 95% CI [1.76, 1.99]; $p < 0.001$) (Fig 3). The RRs between infections and dementia were lower: 1.22 (95% CI [1.17, 1.27]; $p < 0.001$) for cystitis and 1.21 (95% CI [1.16, 1.28]; $p < 0.001$) for bacterial

**Table 1. Characteristics of the study population in the main analyses of late-onset dementia.**

| | Dementia | No dementia |
|---|---|---|
| | **N = 62,555** | **N = 312,772** |
| Age (at dementia or index year) | 81.0 (7.0) | 81.0 (7.0) |
| Age (at the start of the exposure period) | 60.0 (7.0) | 60.0 (7.0) |
| Women | 37,582 (60.1%) | 187,910 (60.1%) |
| Education (at the start of the exposure period) | | |
| High | 10,959 (17.5%) | 61,698 (19.7%) |
| Medium | 14,554 (23.3%) | 73,249 (23.4%) |
| Basic | 37,042 (59.2%) | 177,825 (56.9%) |
| Marital status (at the start of the exposure period) | | |
| Never married | 5,948 (9.5%) | 26,854 (8.6%) |
| Married | 41,904 (67.0%) | 217,375 (69.5%) |
| Divorced | 7,803 (12.5%) | 35,135 (11.2%) |
| Widowed | 6,900 (11.0%) | 33,408 (10.7%) |
| Employment status (at the start of the exposure period) | | |
| Aged <60 and employed | 17,335 (27.7%) | 94,662 (30.3%) |
| Aged <60 and retired | 5,381 (8.6%) | 22,260 (7.1%) |
| Aged <60 and other nonworking | 6,334 (10.1%) | 28,328 (9.1%) |
| Aged 60+ | 33,505 (53.6%) | 167,522 (53.6%) |

Data are N (%) or mean (SD).

infection of unspecified site. The diseases were diagnosed on average 4.7–11.8 years before dementia diagnosis, with depression diagnosed the earliest (mean 11.8 [SD 6.2] years before dementia) and organic mental disorder the latest (mean 4.7 [SD 3.8] years). Cystitis and bacterial infection of unspecified site occurred on average 6.5 (SD 5.0) years and 5.6 (SD 3.8) years before dementia, respectively.

In the second part of the analysis, we constructed disease trajectories from these 29 dementia-related diseases, forming 406 disease pairs. Considering the more commonly occurring temporal order in each disease pair, 153 pairs (37.7%) showed a RR ≥ 1.20 with a statistically significant $p$-value after Bonferroni correction ($p < 0.00012$). These were used to construct a network of disease trajectories. Fig 4 presents trajectories with particularly strong associations (RR ≥ 3.00); all 153 disease pairs are listed in the appendix (Table B in S1 Appendix). In supplementary analyses examining the less commonly occurring temporal order, 79 (19.5%) of the 406 pairs remained significant (RR ≥ 1.20, $p < 0.00012$) (Table C in S1 Appendix).

The third part of the analysis examined whether the associations between infections and dementia were explained by other dementia-related diseases. Of the 29 dementia-related diseases identified in the first part of the analysis, 19 (65.5%) were associated with an increased risk of cystitis (Fig 5), with the strongest associations for epilepsy (RR 2.69, 95% CI [2.02, 3.59]; $p < 0.001$), intracerebral haemorrhage (RR 2.69, 95% CI [1.88, 3.85]; $p < 0.001$), and alcohol-related mental and behavioural disorders (RR 2.61, 95% CI 1.93, 3.54]; $p < 0.001$). Ten diseases (34.5%) were associated with bacterial infections of unspecified site (Fig 6), most strongly other fluid, electrolyte and acid-base balance disorders (RR 2.57, 95% CI [1.95, 3.40]; $p < 0.001$), retinal disorders in diseases classified elsewhere (RR 2.46, 95% CI [1.70, 3.58]; $p < 0.001$), and alcohol-related mental and behavioural disorders (RR 2.30, 95% CI [1.77, 2.99]; $p < 0.001$).

The associations of infections with dementia risk remained robust after adjustment for all prior noninfectious dementia-related diseases. After these adjustments, the RR was 1.19 (95% CI [1.14, 1.24]; $p < 0.001$) for cystitis and 1.19 (95%

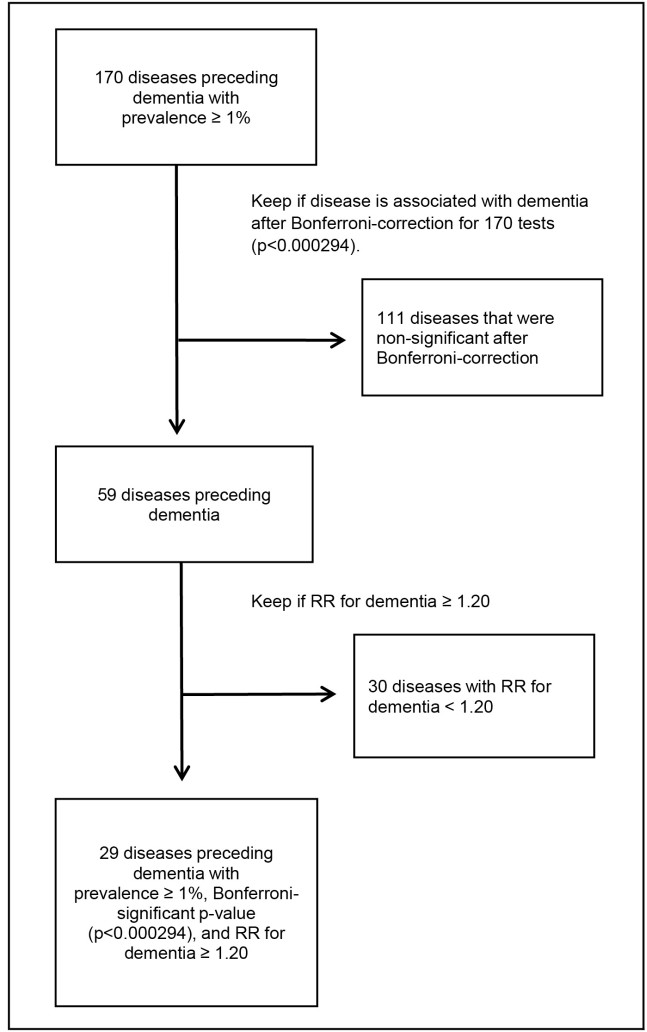

**Fig 2. Flow chart of disease selection.** The figure shows the steps used to identify the 29 diseases robustly associated with incident dementia from 170 diseases with a prevalence of at least 1% before dementia. Abbreviation: RR, rate ratio.

CI [1.13, 1.25]; $p < 0.001$) for bacterial infections of unspecified site. Only 10.8% to 13.8% of the excess dementia risk among individuals with these infections was attributable to comorbidities. The findings were consistent across sex and education strata (Fig 7) and in all sensitivity analyses (Fig A in S1 Appendix). In lag-analyses for cystitis and bacterial infections of unspecified site, the associations with dementia attenuated but remained statistically significant (Fig B in S1 Appendix).

In additional analyses, we considered early onset dementia (diagnosed before age 65). We identified 2,639 cases (mean age 57.5 [SD 5.2], 1,264 [47.9%] women) with early onset dementia and 13,195 control individuals (mean age 57.5 [SD 5.2], 6,320 [47.9%] women) matched for year of birth, sex, and follow-up period (Table D in S1 Appendix). Using a similar stepwise procedure than in the main analysis, we found that after Bonferroni correction for multiple testing (162 tested diseases, $P < 0.000309$), 42 diseases were associated with a RR of at least 1.20 for early onset dementia (Fig 8). 1,665 (63.1%) of the 2,639 cases with early onset dementia had at least one of these diseases during the two-decade exposure period before dementia (Table E in S1 Appendix). Strongest associations with early onset dementia were seen

| Exposure (ICD-10 code and name) | N exposed controls/all controls | % controls exposed | N exposed dementia cases/all dementia cases | % dementia cases exposed | Mean time before dementia, years (SD) | RR (95% CI) | P-value |
|---|---|---|---|---|---|---|---|
| F06 Other oganic mental disorders | 1,508/312,772 | 0.5% | 1,141/62,555 | 1.8% | 4.7 (3.8) | 3.76 (3.48, 4.06) | <0.0001 |
| G20 Parkinson disease | 2,898/312,772 | 0.9% | 1,837/62,555 | 2.9% | 7.1 (4.7) | 3.24 (3.06, 3.44) | <0.0001 |
| F10 Mental and behavioural disorders due to use of alcohol | 4,122/312,772 | 1.3% | 1,598/62,555 | 2.6% | 10.6 (6.1) | 1.87 (1.76, 1.99) | <0.0001 |
| G40 Epilepsy | 3,894/312,772 | 1.2% | 1,233/62,555 | 2.0% | 10.0 (6.2) | 1.56 (1.46, 1.66) | <0.0001 |
| I95 Hypotension | 2,794/312,772 | 0.9% | 863/62,555 | 1.4% | 5.8 (4.5) | 1.52 (1.41, 1.65) | <0.0001 |
| S06 Intracranial injury | 12,545/312,772 | 4.0% | 3,757/62,555 | 6.0% | 7.8 (5.5) | 1.51 (1.45, 1.57) | <0.0001 |
| I67 Other cerebrovascular diseases | 2,133/312,772 | 0.7% | 637/62,555 | 1.0% | 8.6 (5.7) | 1.49 (1.36, 1.63) | <0.0001 |
| I61 Intracerebral haemorrhage | 2,722/312,772 | 0.9% | 806/62,555 | 1.3% | 8.2 (5.5) | 1.49 (1.38, 1.61) | <0.0001 |
| F41 Other anxiety disorders | 3,341/312,772 | 1.1% | 963/62,555 | 1.5% | 10.1 (6.1) | 1.43 (1.33, 1.53) | <0.0001 |
| F32 Depressive episode | 8,636/312,772 | 2.8% | 2,451/62,555 | 3.9% | 11.8 (6.2) | 1.41 (1.35, 1.48) | <0.0001 |
| S01 Open wound of head | 9,746/312,772 | 3.1% | 2,698/62,555 | 4.3% | 7.5 (5.7) | 1.40 (1.34, 1.47) | <0.0001 |
| S02 Fracture of skull and facial bones | 2,359/312,772 | 0.8% | 666/62,555 | 1.1% | 9.3 (5.8) | 1.40 (1.28, 1.53) | <0.0001 |
| I69 Sequelae of cerebrovascular disease | 5,391/312,772 | 1.7% | 1,510/62,555 | 2.4% | 7.7 (5.2) | 1.39 (1.31, 1.48) | <0.0001 |
| G25 Other extrapyramidal and movement disorders | 3,099/312,772 | 1.0% | 854/62,555 | 1.4% | 8.8 (5.2) | 1.37 (1.26, 1.47) | <0.0001 |
| S00 Superficial injury of head | 2,765/312,772 | 0.9% | 765/62,555 | 1.2% | 6.8 (5.2) | 1.36 (1.26, 1.48) | <0.0001 |
| F33 Recurrent depressive disorder | 3,081/312,772 | 1.0% | 845/62,555 | 1.4% | 11.4 (5.7) | 1.35 (1.25, 1.46) | <0.0001 |
| E11 Type 2 diabetes mellitus | 11,440/312,772 | 3.7% | 3,094/62,555 | 4.9% | 10.3 (5.7) | 1.32 (1.27, 1.38) | <0.0001 |
| H36 Retinal disorders in diseases classified elsewhere | 3,015/312,772 | 1.0% | 800/62,555 | 1.3% | 10.9 (5.6) | 1.29 (1.20, 1.40) | <0.0001 |
| S70 Superficial injury of hip and thigh | 3,463/312,772 | 1.1% | 902/62,555 | 1.4% | 7.0 (5.3) | 1.29 (1.20, 1.39) | <0.0001 |
| E87 Other disorders of fluid, electrolyte and acid-base balance | 6,491/312,772 | 2.1% | 1,656/62,555 | 2.6% | 6.6 (4.5) | 1.26 (1.19, 1.33) | <0.0001 |
| S72 Fracture of femur | 11,887/312,772 | 3.8% | 2,959/62,555 | 4.7% | 6.8 (4.9) | 1.25 (1.20, 1.30) | <0.0001 |
| I63 Cerebral infarction | 24,402/312,772 | 7.8% | 5,989/62,555 | 9.6% | 8.6 (5.2) | 1.24 (1.20, 1.27) | <0.0001 |
| S32 Fracture of lumbar spine and pelvis | 5,942/312,772 | 1.9% | 1,473/62,555 | 2.4% | 6.9 (5.1) | 1.23 (1.16, 1.30) | <0.0001 |
| F43 Reaction to severe stress, and adjustment disorders | 2,616/312,772 | 0.8% | 636/62,555 | 1.0% | 11.3 (6.0) | 1.22 (1.12, 1.33) | <0.0001 |
| N30 Cystitis | 12,024/312,772 | 3.8% | 2,923/62,555 | 4.7% | 6.5 (5.0) | 1.22 (1.17, 1.27) | <0.0001 |
| A49 Bacterial infection of unspecified site | 8,401/312,772 | 2.7% | 2,040/62,555 | 3.3% | 5.6 (3.8) | 1.21 (1.16, 1.28) | <0.0001 |
| K59 Other functional intestinal disorders | 10,490/312,772 | 3.4% | 2,557/62,555 | 4.1% | 7.8 (5.2) | 1.21 (1.16, 1.27) | <0.0001 |
| S22 Fracture of rib(s), sternum and thoracic spine | 5,704/312,772 | 1.8% | 1,387/62,555 | 2.2% | 8.2 (5.7) | 1.21 (1.14, 1.28) | <0.0001 |
| S42 Fracture of shoulder and upper arm | 9,596/312,772 | 3.1% | 2,324/62,555 | 3.7% | 9.0 (5.6) | 1.21 (1.15, 1.27) | <0.0001 |

**Fig 3. Diseases associated with an increased risk of dementia.** Rate ratios and *p*-values are based on incidence-density sampling and matched analysis (conditional logistic regression) and were adjusted for year of birth (matching variable), index date (matching variable), sex (matching variable), education, marital status, area of residence, and employment status. Abbreviations: CI, confidence interval; Other organic mental disorders, Other mental disorders due to brain damage and dysfunction and to physical disease. Abbreviation: RR, rate ratio.

**Table 2. Number of dementia cases and controls with dementia-related infections for late-onset dementia and early onset dementia.**

| | Late-onset dementia | No late-onset dementia | |
|---|---|---|---|
| | *N*=62,555 | *N*=312,772 | *p* for difference |
| Cystitis | 2,923 (4.7%) | 12,024 (3.8%) | <0.001 |
| Bacterial infection of unspecified site | 2,040 (3.3%) | 8,401 (2.7%) | <0.001 |
| | **Early onset dementia** | **No early onset dementia** | |
| | *N*=2,639 | *N*=13,195 | *p* for difference |
| Other gastroenteritis and colitis of infectious and unspecified origin | 105 (4.0%) | 270 (2.0%) | <0.001 |
| Bacterial infection of unspecified site | 70 (2.7%) | 131 (1.0%) | <0.001 |
| Bacterial pneumonia, not elsewhere classified | 74 (2,8%) | 181 (1.4%) | <0.001 |
| Pneumonia, organism unspecified | 192 (7.3%) | 438 (3.3%) | <0.001 |
| Dental caries | 76 (2.9%) | 111 (0.8%) | <0.001 |

Data are N (%) or mean (SD). *P*-values were computed using the chi-squared test.

for Parkinson's disease (RR 31.88, 95% CI [19.50, 52.11]; *p* < 0.001), sequalae of injuries of head (RR 10.23, 95% CI [6.17, 16.97]; *p* < 0.001), and other mental disorders due to brain damage and dysfunction and to physical disease (RR 10.11, 95% CI [6.37, 16.03]; *p* < 0.001).

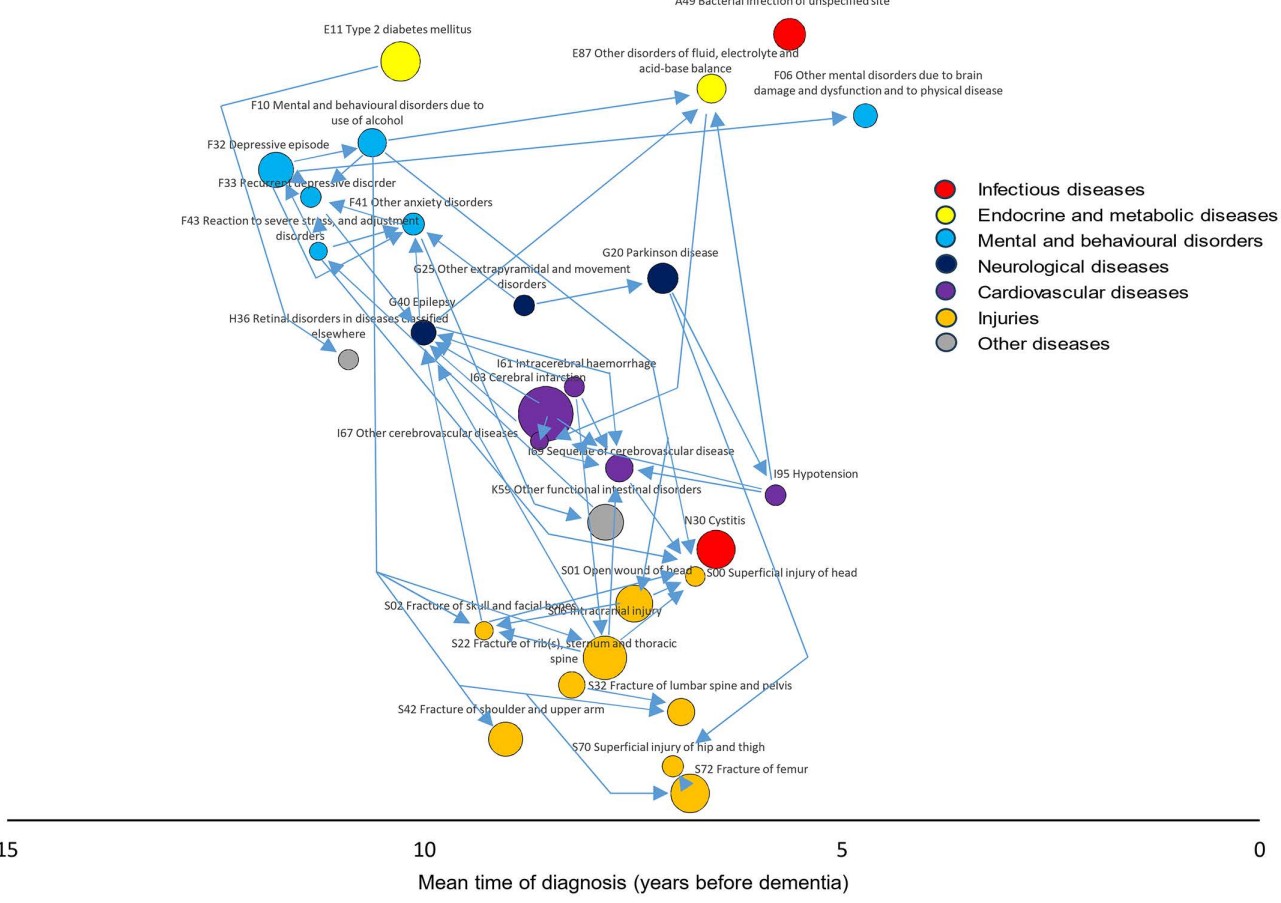

**Fig 4. Associations between dementia-related diseases.** The arrows indicate disease pairs in which the first-occurring diseases (tail of the arrow) is associated with a Bonferroni significant RR of at least 3.00 for the second-occurring disease (arrowhead). RRs and *p*-values are based on incidence-density sampling and matched analysis (conditional logistic regression) and were adjusted for year of birth (matching variable), index date (matching variable), sex (matching variable), education, marital status, area of residence, and employment status. Abbreviation: RR, rate ratio.

Of infections, other gastroenteritis and colitis of infectious and unspecified origin had a RR of 1.88 (95% CI [1.48, 2.39]; *p* < 0.001), bacterial infection of unspecified site had a RR of 2.24 (95% CI [1.64, 3.06]; *p* < 0.001), bacterial pneumonia, not elsewhere classified a RR of 1.75 (95% CI [1.31, 2.34]; *p* < 0.001), pneumonia, organism unspecified a RR of 1.82 (95% CI [1.50, 2.19]; *p* < 0.001), and dental caries a RR of 2.20 (95% CI [1.59, 3.03]; *p* < 0.001) for early onset dementia. The associations of these infections with an increased risk of early onset dementia remained after adjustment for all prior noninfectious early onset dementia-related diseases (Fig 9). Of the excess risk of early onset dementia among individuals with these infections, 8.8% to 32.8% was attributable to comorbidities.

## Discussion

In this analysis of electronic health records from nationwide registries, we examined 170 physical and mental diseases among more than 60,000 cases of late-onset dementia (age 65+) and over 300,000 matched controls. We identified 29 diseases that were robustly associated with an increased risk of dementia and identified disease trajectories in which the onset of one condition elevated the risk of developing another. Two of these diseases were severe infections—hospital-treated cystitis and bacterial infections of unspecified site—typically occurring late in the disease trajectories.

### Association of diseases with N30 Cystitis

| Exposure | RR (95% CI) |
| --- | --- |
| A49 Bacterial infection of unspecified site | 2.30 (1.79, 2.95) |
| E11 Type 2 diabetes mellitus | 1.99 (1.65, 2.39) |
| E87 Other disorders of fluid, electrolyte and acid-base balance | 2.42 (1.91, 3.06) |
| F10 Mental and behavioural disorders due to use of alcohol | 2.61 (1.93, 3.54) |
| F32 Depressive episode | 2.13 (1.76, 2.57) |
| F33 Recurrent depressive disorder | 2.37 (1.74, 3.22) |
| F41 Other anxiety disorders | 2.01 (1.49, 2.72) |
| F43 Reaction to severe stress, and adjustment disorders | 2.28 (1.57, 3.31) |
| G20 Parkinson disease | 2.27 (1.72, 3.00) |
| G40 Epilepsy | 2.69 (2.02, 3.59) |
| I61 Intracerebral haemorrhage | 2.69 (1.88, 3.85) |
| I63 Cerebral infarction | 1.85 (1.61, 2.13) |
| I69 Sequelae of cerebrovascular disease | 2.00 (1.51, 2.65) |
| K59 Other functional intestinal disorders | 2.59 (2.17, 3.10) |
| S06 Intracranial injury | 1.84 (1.54, 2.19) |
| S22 Fracture of rib(s), sternum and thoracic spine | 1.80 (1.35, 2.39) |
| S42 Fracture of shoulder and upper arm | 1.53 (1.25, 1.88) |
| S70 Superficial injury of hip and thigh | 2.16 (1.58, 2.94) |
| S72 Fracture of femur | 1.56 (1.29, 1.89) |

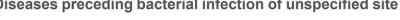

**Fig 5. Diseases associated with an increased risk of cystitis.** The arrows indicate disease pairs in which the first-occurring diseases (tail of the arrow) are associated with a Bonferroni significant RR of at least 1.20 for cystitis (arrowhead). RRs and *p*-values are based on incidence-density sampling and matched analysis (conditional logistic regression) and were adjusted for year of birth (matching variable), index date (matching variable), sex (matching variable), education, marital status, area of residence, and employment status. Abbreviations: CI, confidence interval; RR, rate ratio.

### Association of diseases with A49 Bacterial infection of unspecified site

| Exposure | RR (95% CI) |
| --- | --- |
| E11 Type 2 diabetes mellitus | 2.07 (1.70, 2.52) |
| E87 Other disorders of fluid, electrolyte and acid-base balance | 2.57 (1.95, 3.40) |
| F10 Mental and behavioural disorders due to use of alcohol | 2.30 (1.77, 2.99) |
| F33 Recurrent depressive disorder | 2.14 (1.47, 3.11) |
| G40 Epilepsy | 2.05 (1.51, 2.79) |
| H36 Retinal disorders in diseases classified elsewhere | 2.46 (1.70, 3.58) |
| I63 Cerebral infarction | 1.56 (1.32, 1.84) |
| I69 Sequelae of cerebrovascular disease | 1.79 (1.33, 2.42) |
| S06 Intracranial injury | 1.64 (1.32, 2.04) |
| S72 Fracture of femur | 1.89 (1.50, 2.37) |

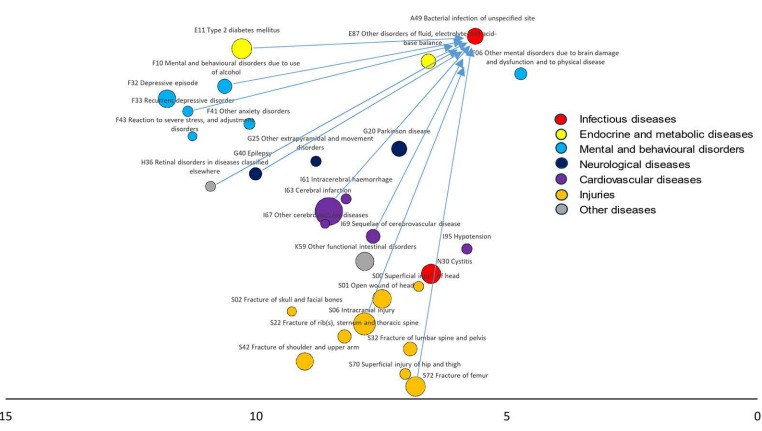

**Fig 6. Diseases associated with an increased risk of bacterial infection of an unspecified site.** The arrows indicate disease pairs in which the first-occurring diseases (tail of the arrow) are associated with a Bonferroni significant RR of at least 1.20 for bacterial infection of an unspecified site (arrowhead). RRs and *p*-values are based on incidence-density sampling and matched analysis (conditional logistic regression) and were adjusted for year of birth (matching variable), index date (matching variable), sex (matching variable), education, marital status, area of residence, and employment status. Abbreviations: CI, confidence interval; RR, rate ratio.

Consequently, many other dementia-related diseases were also associated with an increased risk of these infections. However, as less than one-seventh of the excess dementia risk among individuals with hospital-treated cystitis or bacterial infections of unspecified site was attributable to comorbidities, the present study suggests that the associations between

| Exposure (ICD-10 code and name) | N exposed controls/all controls | % controls exposed | N exposed dementia cases/all dementia cases | % dementia cases exposed | RR (95% CI) | P-value | ERE |
|---|---|---|---|---|---|---|---|
| **N30 Cystitis** | | | | | | | |
| ALL PARTICIPANTS | | | | | | | |
| Adjusted model | 12,024/312,772 | 3.8% | 2,923/62,555 | 4.7% | 1.22 (1.17, 1.27) | <0.0001 | |
| Adjusted model + additionally adjusted for other diseases | 12,024/312,772 | 3.8% | 2,923/62,555 | 4.7% | 1.19 (1.14, 1.24) | <0.0001 | 13.8% |
| MEN | | | | | | | |
| Adjusted model | 2,189/124,862 | 1.8% | 553/24,973 | 2.2% | 1.25 (1.13, 1.37) | <0.0001 | |
| Adjusted model + additionally adjusted for other diseases | 2,189/124,862 | 1.8% | 553/24,973 | 2.2% | 1.21 (1.10, 1.34) | <0.0001 | 13.0% |
| WOMEN | | | | | | | |
| Adjusted model | 9,835/187,910 | 5.2% | 2,370/37,582 | 6.3% | 1.21 (1.16, 1.27) | <0.0001 | |
| Adjusted model + additionally adjusted for other diseases | 9,835/187,910 | 5.2% | 2,370/37,582 | 6.3% | 1.19 (1.13, 1.24) | <0.0001 | 12.6% |
| HIGH EDUCATION | | | | | | | |
| Adjusted model | 4,096/134,947 | 3.0% | 982/25,513 | 3.8% | 1.35 (1.24, 1.48) | <0.0001 | |
| Adjusted model + additionally adjusted for other diseases | 4,096/134,947 | 3.0% | 982/25,513 | 3.8% | 1.31 (1.20, 1.43) | 0.0001 | 11.7% |
| LOW EDUCATION | | | | | | | |
| Adjusted model | 7,928/177,825 | 4.5% | 1,941/37,042 | 5.2% | 1.18 (1.12, 1.25) | <0.0001 | |
| Adjusted model + additionally adjusted for other diseases | 7,928/177,825 | 4.5% | 1,941/37,042 | 5.2% | 1.17 (1.10, 1.23) | <0.0001 | 7.8% |
| **A49 Bacterial infection of unspecified site** | | | | | | | |
| ALL PARTICIPANTS | | | | | | | |
| Adjusted model | 8,401/312,772 | 2.7% | 2,040/62,555 | 3.3% | 1.21 (1.16, 1.28) | <0.0001 | |
| Adjusted model + additionally adjusted for other diseases | 8,401/312,772 | 2.7% | 2,040/62,555 | 3.3% | 1.19 (1.13, 1.25) | <0.0001 | 10.8% |
| MEN | | | | | | | |
| Adjusted model | 3,611/124,862 | 2.9% | 930/24,973 | 3.7% | 1.29 (1.20, 1.39) | <0.0001 | |
| Adjusted model + additionally adjusted for other diseases | 3,611/124,862 | 2.9% | 930/24,973 | 3.7% | 1.26 (1.16, 1.35) | <0.0001 | 13.0% |
| WOMEN | | | | | | | |
| Adjusted model | 4,790/187,910 | 2.5% | 1,110/37,582 | 3.0% | 1.16 (1.08, 1.23) | <0.0001 | |
| Adjusted model + additionally adjusted for other diseases | 4,790/187,910 | 2.5% | 1,110/37,582 | 3.0% | 1.14 (1.07, 1.22) | 0.0001 | 8.2% |
| HIGH EDUCATION | | | | | | | |
| Adjusted model | 3,189/134,947 | 2.4% | 752/25,513 | 2.9% | 1.25 (1.13, 1.38) | <0.0001 | |
| Adjusted model + additionally adjusted for other diseases | 3,189/134,947 | 2.4% | 752/25,513 | 2.9% | 1.21 (1.10, 1.34) | 0.0001 | 13.1% |
| LOW EDUCATION | | | | | | | |
| Adjusted model | 5,212/177,825 | 2.9% | 1,288/37,042 | 3.5% | 1.21 (1.13, 1.30) | <0.0001 | |
| Adjusted model + additionally adjusted for other diseases | 5,212/177,825 | 2.9% | 1,288/37,042 | 3.5% | 1.20 (1.12, 1.28) | <0.0001 | 5.7% |

Rate ratio (95% CI): 0.5 — 1 — 2

**Fig 7. Association of infectious diseases with the risk of dementia in the whole study sample and in subsamples stratified by sex and education.** Rate ratios and *p*-values are based on incidence-density sampling and matched analysis (conditional logistic regression) and were adjusted for year of birth (matching variable), index date (matching variable), sex (matching variable), education, marital status, area of residence, and employment status. Analyses stratified by education were not adjusted for education. Additionally, adjusted models were also adjusted for all noninfectious dementia-related diseases identified in this study. Abbreviations: CI, confidence interval; ERE, excess risk explained by comorbidities; RR, rate ratio.

these infections and dementia are largely independent of prior conditions and thus supports the idea that severe infections are risk factors for dementia.

During the past few years, considerable research interest has focussed on the potential role of infectious diseases in the development of dementia [1–6,30–32]. Pre-existing comorbid conditions are potentially important contributors to this association, because the average age of dementia diagnosis is older than 80 years, and therefore people with dementia often have other illnesses [12,13], many of which also confer an increased risk for infections [16–18]. Supporting this reasoning, we found that of the 29 diseases associated with an increased risk of dementia, two-thirds were also linked to an increased risk of cystitis, and one third to an increased risk of bacterial infections of unspecified site. These included known infection risk factors, such as type 2 diabetes, alcohol abuse, cerebral infarction, Parkinson's disease, epilepsy, and traumatic brain injury [16,17,19,20]. However, they also included conditions that are not typically considered risk factors for infections, such as nonbrain injuries and depression.

| Exposure (ICD-10 code and name) | N exposed controls/all controls | % controls exposed | N exposed dementia cases/all dementia cases | % dementia cases exposed | Mean time before dementia, years (SD) | RR (95% CI) | P-value |
|---|---|---|---|---|---|---|---|
| G20 Parkinson disease | 24/13,195 | 0.2% | 104/2,639 | 3.9% | 9.0 (5.2) | 31.88 (19.50, 52.11) | <0.0001 |
| T90 Sequelae of injuries of head | 24/13,195 | 0.2% | 50/2,639 | 1.9% | 8.2 (5.9) | 10.23 (6.17, 16.97) | <0.0001 |
| F06 Other organic mental disorders | 29/13,195 | 0.2% | 58/2,639 | 2.2% | 5.9 (4.5) | 10.11 (6.37, 16.03) | <0.0001 |
| E87 Other disorders of fluid, electrolyte and acid-base balance | 68/13,195 | 0.5% | 73/2,639 | 2.8% | 7.1 (4.3) | 4.64 (3.25, 6.65) | <0.0001 |
| G35 Multiple sclerosis | 51/13,195 | 0.4% | 47/2,639 | 1.8% | 13.6 (5.8) | 4.25 (2.80, 6.45) | <0.0001 |
| G25 Other extrapyramidal and movement disorders | 46/13,195 | 0.3% | 36/2,639 | 1.4% | 9.0 (5.2) | 4.16 (2.62, 6.61) | <0.0001 |
| G40 Epilepsy | 162/13,195 | 1.2% | 161/2,639 | 6.1% | 10.9 (6.5) | 3.73 (2.93, 4.75) | <0.0001 |
| F29 Unspecified nonorganic psychosis | 99/13,195 | 0.8% | 82/2,639 | 3.1% | 9.8 (6.1) | 3.46 (2.54, 4.73) | <0.0001 |
| I69 Sequelae of cerebrovascular disease | 81/13,195 | 0.6% | 52/2,639 | 2.0% | 7.3 (5.7) | 3.28 (2.27, 4.74) | <0.0001 |
| F10 Mental and behavioural disorders due to use of alcohol | 474/13,195 | 3.6% | 289/2,639 | 11.0% | 10.4 (5.8) | 3.07 (2.61, 3.62) | <0.0001 |
| K70 Alcoholic liver disease | 47/13,195 | 0.4% | 29/2,639 | 1.1% | 9.0 (5.1) | 3.06 (1.88, 4.98) | <0.0001 |
| I67 Other cerebrovascular diseases | 50/13,195 | 0.4% | 31/2,639 | 1.2% | 8.7 (6.4) | 3.05 (1.91, 4.85) | <0.0001 |
| F31 Bipolar affective disorder | 142/13,195 | 1.1% | 88/2,639 | 3.3% | 11.1 (5.2) | 2.97 (2.24, 3.94) | <0.0001 |
| H36 Retinal disorders in diseases classified elsewhere | 76/13,195 | 0.6% | 44/2,639 | 1.7% | 10.5 (6.7) | 2.81 (1.90, 4.16) | <0.0001 |
| S06 Intracranial injury | 356/13,195 | 2.7% | 191/2,639 | 7.2% | 9.2 (6.0) | 2.60 (2.15, 3.15) | <0.0001 |
| K92 Other diseases of digestive system | 104/13,195 | 0.8% | 61/2,639 | 2.3% | 9.7 (5.3) | 2.59 (1.85, 3.61) | <0.0001 |
| E10 Type 1 diabetes mellitus | 128/13,195 | 1.0% | 65/2,639 | 2.5% | 12.6 (6.2) | 2.59 (1.89, 3.54) | <0.0001 |
| I50 Heart failure | 71/13,195 | 0.5% | 38/2,639 | 1.4% | 6.0 (3.9) | 2.49 (1.65, 3.78) | <0.0001 |
| I63 Cerebral infarction | 191/13,195 | 1.4% | 99/2,639 | 3.8% | 8.2 (5.2) | 2.45 (1.88, 3.18) | <0.0001 |
| F99 Mental disorder, not otherwise specified | 73/13,195 | 0.6% | 38/2,639 | 1.4% | 9.6 (6.1) | 2.44 (1.61, 3.69) | <0.0001 |
| D64 Other anaemias | 95/13,195 | 0.7% | 52/2,639 | 2.0% | 8.4 (5.7) | 2.33 (1.63, 3.34) | <0.0001 |
| S02 Fracture of skull and facial bones | 161/13,195 | 1.2% | 74/2,639 | 2.8% | 12.0 (5.5) | 2.31 (1.73, 3.09) | <0.0001 |
| K59 Other functional intestinal disorders | 138/13,195 | 1.0% | 75/2,639 | 2.8% | 8.2 (5.6) | 2.27 (1.67, 3.09) | <0.0001 |
| A49 Bacterial infection of unspecified site | 131/13,195 | 1.0% | 70/2,639 | 2.7% | 6.7 (4.3) | 2.24 (1.64, 3.06) | <0.0001 |
| K02 Dental caries | 111/13,195 | 0.8% | 76/2,639 | 2.9% | 10.8 (6.2) | 2.20 (1.59, 3.03) | <0.0001 |
| T36 Poisoning by systemic antibiotics | 193/13,195 | 1.5% | 101/2,639 | 3.8% | 11.9 (5.7) | 2.18 (1.69, 2.83) | <0.0001 |
| D50 Iron deficiency anaemia | 109/13,195 | 0.8% | 54/2,639 | 2.0% | 8.6 (5.2) | 2.15 (1.52, 3.04) | <0.0001 |
| F32 Depressive episode | 767/13,195 | 5.8% | 316/2,639 | 12.0% | 11.1 (5.9) | 2.15 (1.86, 2.48) | <0.0001 |
| S00 Superficial injury of head | 121/13,195 | 0.9% | 57/2,639 | 2.2% | 10.6 (6.6) | 2.14 (1.53, 2.98) | <0.0001 |
| S01 Open wound of head | 392/13,195 | 3.0% | 170/2,639 | 6.4% | 10.3 (6.3) | 2.12 (1.75, 2.57) | <0.0001 |
| F41 Other anxiety disorders | 314/13,195 | 2.4% | 132/2,639 | 5.0% | 11.1 (6.2) | 2.12 (1.71, 2.63) | <0.0001 |
| S42 Fracture of shoulder and upper arm | 246/13,195 | 1.9% | 102/2,639 | 3.9% | 9.3 (6.2) | 1.94 (1.51, 2.49) | <0.0001 |
| A09 Other gastroenteritis and colitis of infectious and unspecified origin | 270/13,195 | 2.0% | 105/2,639 | 4.0% | 9.2 (5.5) | 1.88 (1.48, 2.39) | <0.0001 |
| F33 Recurrent depressive disorder | 395/13,195 | 3.0% | 148/2,639 | 5.6% | 10.3 (5.5) | 1.87 (1.53, 2.28) | <0.0001 |
| J18 Pneumonia, organism unspecified | 438/13,195 | 3.3% | 192/2,639 | 7.3% | 7.5 (4.6) | 1.82 (1.50, 2.19) | <0.0001 |
| G45 Transient cerebral ischaemic attacks and related syndromes | 241/13,195 | 1.8% | 91/2,639 | 3.4% | 6.9 (4.6) | 1.81 (1.40, 2.35) | <0.0001 |
| F43 Reaction to severe stress, and adjustment disorders | 261/13,195 | 2.0% | 92/2,639 | 3.5% | 9.4 (5.7) | 1.79 (1.40, 2.30) | <0.0001 |
| H53 Visual disturbances | 206/13,195 | 1.6% | 70/2,639 | 2.7% | 9.7 (5.9) | 1.77 (1.33, 2.34) | 0.0001 |
| J15 Bacterial pneumonia, not elsewhere classified | 181/13,195 | 1.4% | 74/2,639 | 2.8% | 10.6 (5.3) | 1.75 (1.31, 2.34) | 0.0001 |
| S82 Fracture of lower leg, including ankle | 594/13,195 | 4.5% | 199/2,639 | 7.5% | 11.1 (6.0) | 1.58 (1.33, 1.88) | <0.0001 |
| H90 Conductive and sensorineural hearing loss | 499/13,195 | 3.8% | 149/2,639 | 5.6% | 9.9 (5.8) | 1.49 (1.22, 1.81) | 0.0001 |
| M54 Dorsalgia | 1,164/13,195 | 8.8% | 303/2,639 | 11.5% | 10.1 (5.6) | 1.31 (1.14, 1.51) | <0.0001 |

Rate ratio (95% CI): 0.5  1  2  4  8  16  32

**Fig 8. Diseases associated with an increased risk of early onset dementia.** Rate ratios and *p*-values are based on incidence-density sampling and matched analysis (conditional logistic regression) and were adjusted for year of birth (matching variable), index date (matching variable), sex (matching variable), education, marital status, area of residence, and employment status. Abbreviations: CI, confidence interval; Other organic mental disorders, Other mental disorders due to brain damage and dysfunction and to physical disease; RR, rate ratio.

The modest contribution of pre-existing conditions to the association between infections and dementia supports the hypothesis that severe infections may directly influence dementia risk. In our study, the average time difference between infection diagnosis and dementia was relatively short, 5–6 years. This time frame suggests that the inflammatory insult resulting from infections severe enough to require hospital treatment may accelerate pre-existing preclinical stage of dementia rather than initiate neurodegeneration in a cognitively healthy person [31,33,34].

Evidence from cohort studies and natural experiments indicates that vaccination against infectious diseases might be an effective strategy for reducing or postponing the onset of dementia [35–38]. In these natural experiments, herpes zoster vaccination reduced dementia risk more effectively among women than men [37,38]. Conversely, in our data focussed on other infections, cystitis and bacterial infections of unspecified site both had slightly stronger associations with dementia among men. Urinary tract and bacterial infections are common infectious diseases that have been associated with increased dementia risk in several prospective studies [3–6,32], but findings stratified by sex have been mixed. In a large British study, the association between urinary tract infections and dementia was stronger among men [3], but in a large

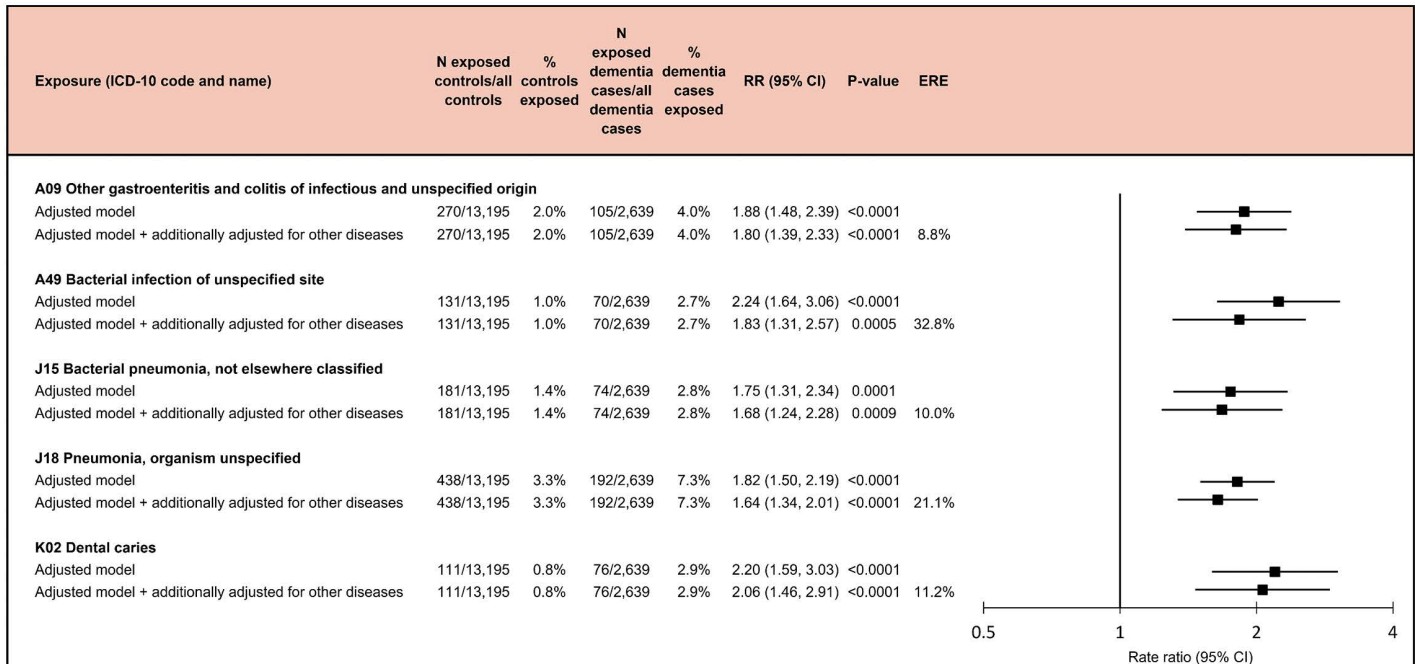

**Fig 9. Infectious diseases associated with an increased risk of early onset dementia.** Rate ratios and *p*-values are based on incidence-density sampling and matched analysis (conditional logistic regression) and were adjusted for year of birth (matching variable), index date (matching variable), sex (matching variable), education, marital status, area of residence, and employment status. Additionally, adjusted models were also adjusted for all noninfectious early onset dementia-related diseases identified in this study. Abbreviations: CI, confidence interval; ERE, excess risk explained by comorbidities; RR, rate ratio.

New Zealand study the association between bacterial infections and dementia was stronger among women [5]. Also, somewhat surprisingly, in our study, the associations of these infections with dementia were stronger among those with high education compared to those with low education. We speculate that the relative contribution of infections on dementia risk might be stronger among highly educated individuals because they have a lower prevalence of other dementia risk factors [39]. Consistent with previous findings, infections showed stronger associations with early onset than with late-onset dementia and we identified more dementia-associated infections in early onset dementia [4]. The mechanisms underlying these differences are unclear, but it is well-established that early and late-onset dementia differ in aetiology and genetic susceptibility [40,41].

As pertains to noninfectious diseases, our findings agree with previous studies suggesting increased risk of dementia among individuals with Parkinson's disease, mental and behavioural disorders due to use of alcohol, epilepsy, intracranial injury, ischaemic and haemorrhagic stroke, depression, type 2 diabetes, or retinal disorders in diseases including diabetic retinopathy [42–51]. In contrast, evidence on the status of anxiety as a risk factor remains mixed [52–54]. Among the dementia-related diseases identified in our study, several were related to fractures and other injuries. Increased vulnerability to accidents may result from nervous system deterioration, frailty, or, in some cases, harmful alcohol use [55–59]. Additionally, head injuries are often accompanied by brain trauma [60,61].

We also found that disorders of the fluid, electrolyte and acid-base balance were associated with an increased risk of dementia. These disorders can indicate poor overall health and may also have direct adverse effects on brain health [62–68]. Furthermore, the disease trajectories formed by dementia-related conditions suggest that the cumulative burden of multiple diseases may contribute to dementia risk.

Our study has some important strengths. Combined with correction for multiple testing, the large sample drawn from the total population of Finland provides good statistical power and stable estimates, making chance findings unlikely. Our hypothesis-free approach, estimating associations of dementia with all nonrare diseases (prevalence ≥ 1%) recorded during a 20-year time window before dementia diagnosis, is also a strength, allowing us to identify all relevant diseases regardless of whether their importance has previously been appreciated. The register-based data do not suffer from self-report biases.

Our study also has some important limitations. As this is a population-wide register-based study, baseline cognitive assessments and clinical examination data leading to dementia and other disease diagnoses were not available. However, previous evidence indicates that electronic health record-based diagnoses have acceptable validity for examining disease risk factors; the Finnish Care Register for Health Care includes over 95% of all hospital discharges in Finland with high positive predictive values ranging from 75% to 99% for common diagnoses [69]. We assumed that all study individuals remained covered by the Finnish healthcare system throughout the observation period, thereby ignoring any bias arising from individuals who emigrated from Finland. However, this bias is unlikely to be major, as emigration among Finnish citizens aged 65 years and older is rare—~0.05% annually [21]. Infections were treated in hospitals, but no data on specific treatments were available from the electronic health records. We examined all-cause dementia as the outcome. Future work should investigate dementia subtypes, given that severe infections may be more strongly linked to vascular dementia than to Alzheimer's disease [1,3]. We may have missed some true associations because we analysed each 3-digit diagnostic code separately, ignoring partial overlap between codes, and applied Bonferroni correction for multiple testing, which may be overly conservative when tests (i.e., diseases) are not fully independent. To reduce bias from delays in the recording of dementia in the registers, we used a 1-year gap in analyses, excluding any diagnoses recorded within 1 year before dementia diagnosis. Some dementia cases may have been misclassified as controls owing to delayed dementia diagnosis; this would likely attenuate the observed associations. Given the long preclinical phase of dementia, part of the observed associations might reflect preclinical dementia increasing susceptibility to infectious and other diseases, rather than the reverse, potentially leading to an overestimation of associations. We observed attenuation of the associations in lag-analyses, suggesting that reverse causation might be present or that severe infections may accelerate existing cognitive decline and neuropathological processes [33,34]. Furthermore, although we had access to relevant demographic data such as age, sex, education, employment, marital status, and area of residence, our data lacked information about other possibly relevant covariates such as smoking status and apolipoprotein E genotype. Nondisease events, such as symptoms without a specific diagnosis, were outside the scope of the study, but might still contribute to dementia risk [54].

In conclusion, findings from this nationwide population-based Finnish study indicate that multiple interconnected diseases are associated with an increased risk of dementia. Among them, severe infections, despite their relationships to several other dementia-related diseases, were independently associated with an increased dementia incidence. Overall, our findings support the possibility that severe infections increase dementia risk; however, intervention studies are required to establish whether preventing or effectively treating infections yields benefits for dementia prevention.

## Supporting information

**S1 Study Plan. Pre-specified study plan.**
(PDF)

**S1 RECORD Checklist. The Reporting of Studies Conducted using Observational Routinely-Collected Data (RECORD) guideline checklist.** The checklist is protected under Creative Commons Attribution (CC BY) license.
(PDF)

**S1 Appendix. Supplementary tables and figures.**
(XLSX)

**S1 Stata Codes. Statistical codes used in analysis.**

(TXT)

## Acknowledgments

Disclaimer: The study does not necessarily reflect the European Commission's views and in no way anticipates the Commission's future policy in this area.

## Author contributions

**Conceptualization:** Pyry N. Sipilä, Kaarina Korhonen, Mika Kivimäki, Pekka Martikainen.

**Data curation:** Pyry N. Sipilä, Pekka Martikainen.

**Formal analysis:** Pyry N. Sipilä.

**Funding acquisition:** Pyry N. Sipilä, Joni V. Lindbohm, Mika Kivimäki, Pekka Martikainen.

**Investigation:** Pyry N. Sipilä, Kaarina Korhonen, Joni V. Lindbohm, Mika Kivimäki, Pekka Martikainen.

**Methodology:** Pyry N. Sipilä, Kaarina Korhonen, Joni V. Lindbohm, Mika Kivimäki, Pekka Martikainen.

**Project administration:** Pyry N. Sipilä, Joni V. Lindbohm, Mika Kivimäki, Pekka Martikainen.

**Resources:** Joni V. Lindbohm, Pekka Martikainen.

**Supervision:** Joni V. Lindbohm, Mika Kivimäki, Pekka Martikainen.

**Visualization:** Pyry N. Sipilä, Joni V. Lindbohm.

**Writing – original draft:** Pyry N. Sipilä.

**Writing – review & editing:** Pyry N. Sipilä, Kaarina Korhonen, Joni V. Lindbohm, Mika Kivimäki, Pekka Martikainen.

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
