## [Editor Report · Decision Letter 0]

17 Jun 2025

Dear Dr Sipilä,

Thank you for submitting your manuscript entitled "Severe infections and risk of dementia in Finland: The role of comorbidities in a nation-wide prospective study" for consideration by PLOS Medicine.

Your manuscript has now been evaluated by the PLOS Medicine editorial staff as well as by an academic editor with relevant expertise and I am writing to let you know that we would like to send your submission out for external peer review.

For clinical studies, please upload a copy of your trial study protocol as a supporting information file. The study protocol should be the version submitted for approval to the institutional review board or ethics committee, should include any amendments to the study protocol, as well as the date of their approval by the institutional review or ethics committee. Please also detail any deviations from the study protocol in the Methods section of your manuscript. The editors will consider the protocol and study conduct prior to a final decision for external review.

Please re-submit your manuscript within two working days, i.e. by Jun 19 2025 11:59PM.

Kind regards,

Suzanne De Bruijn, PhD

Associate Editor

PLOS Medicine

---

## [Decision Letter · Decision Letter 1]

17 Oct 2025

Dear Dr Sipilä,

Many thanks for submitting your manuscript "Severe infections and risk of dementia in Finland: The role of comorbidities in a nation-wide prospective study" (PMEDICINE-D-25-02079R1) to PLOS Medicine. I am writing on behalf of my colleague Dr. Suzanne de Bruijn who is presently away. Your paper has been reviewed by subject experts and a statistician; their comments are included below and can also be accessed here: [LINK]

As you will see, the reviewers find the work of potential interest, but have raised a number of important concerns that must be addressed for the editors to assess the advance of the study. After discussing the paper with the editorial team and an academic editor with relevant expertise, I'm pleased to invite you to revise the paper in response to the reviewers' comments.

Please note that the editors require that you clarify to readers in the Abstract the specific advance of these findings over related research, and in the manuscript that you address and acknowledge the possibility of reverse causality (i.e. dementia contributing to risk of severe infection), include data on treatment of infection, or acknowledge the lack as a study limitation, and that you provide a table detailing the numbers of infection types within your cohort in both dementia and early onset dementia cases.

In the instance that all reviewer and editor concerns are addressed, we plan to send the revised paper to some or all of the original reviewers, and we cannot provide any guarantees at this stage regarding publication. Please also be advised that we would seek the input of an additional independent reviewer to consider a revised manuscript.

We ask that you submit your revision by Nov 07 2025 11:59PM. However, if this deadline is not feasible, please contact me by email, and we can discuss a suitable alternative.

Don't hesitate to contact me directly with any questions (sbruijn@plos.org).

Best regards,

Alison

Alison Farrell, PhD

Senior Editor

PLOS Medicine

Comments from the reviewers:

Reviewer #1: "Severe infections and risk of dementia in Finland: The role of comorbidities in a nationwide prospective study" attempts to disentangle the effects of other comorbidities, from the correlation between (severe) infections and dementia incidence. The study included over 370,000 demographic-matched individuals aged 65 years or older in Finland, and adjusted rate ratio analysis suggests cystitis and bacterial infections of an unspecified site as valid risk factors for dementia independent of 27 recorded comorbidities, over 20 years of follow-up.

Strengths of the study include the size and comprehensiveness of the population, which includes all known individuals with dementia in a national cohort, over two decades. Some issues might be considered:

1. In the Methods section, it is stated that all individuals in the Finnish population aged 65+ at end-2016 and who were diagnosed with late-onset dementia between 2017 and 2020, were included; up to 5 controls were then sampled for each case. It might first be clarified as to whether a sufficiently large exposure window (up to 21 years before diagnosis/index date, i.e. from 1987) was available from register data, for all these individuals (e.g. recent immigrants). If not, were such individuals excluded from the analysis, or included with adjustments?

2. The number of controls for each dementia case (up to 5) might be explained. How was the maximum number of matched controls determined (e.g. according to theory, estimated prevalence, etc.)? Why "up to 5" controls, instead of a fixed 5 controls, given that sampling was performed with replacement? Was exact matching required for the three attributes (year of birth, sex, follow-up period)?

3. In the Study design and sampling of controls subsection, it is stated that subjects who died were censored from the risk pool. It might be clarified as to the procedure for subjects who left the health system (e.g. by emigration).

4. Moreover, the coverage of the register data might be discussed further. Would relevant data be missing due to individuals seeking care from private sources?

5. In the Hospitalisations from severe infections or other non-dementia illnesses (exposures) subsection, it is stated that non-disease events were not included in the analysis. Is it thus assumed that such events are uncorrelated to dementia incidence, despite some of them likely to be traumatic and/or chronic? This might be discussed.

6. In the Ascertainment of all-cause dementia cases subsection, both ICD-10 and ICD-9 codes (used prior to 1996) are considered in defining prevalent dementia cases for exclusion. This appears to imply that this exclusion criteria extends to before the two-decade exclusion window. Is there a reason for the differing temporal criteria for comorbidities (up to 1996 only) and prevalent dementia (apparently up to 1987)? This might be briefly clarified.

7. In the Covariates subsection, it is stated that "to account for potential regional differences in health care, we adjusted analyses for area of residence, split into 18 regions". It might be clarified as to how these adjustments were made.

8. In the Statistical analysis subsection, it is stated that only diseases that preceded dementia with a prevalence of at least 1% during the exposure window were analyzed. Was the 1% threshold defined for achieving statistical significance on the cohort size, or other reasons? This might be clarified.

9. In the Statistical analysis subsection, it is stated that in analyzing associations between dementia-related diseases, only the more commonly occurring temporal order was considered for each disease pair. This might be explained further - would not analyzing both orders be more complete, especially since there does not seem a requirement that causality between diseases is unidirectional?

10. In the Statistical analysis subsection, from what could be understood, significant disease pairs were used to construct disease trajectories, before associations of diseases that were infections with dementia, were additionally adjusted for any other dementia-related diseases. Does the analysis of the latter (infections with dementia) rely on the previous analysis of disease pairs? In general, the statistical methodology should be described in greater detail, possibly in supplementary material.

Reviewer #2: The authors investigated the role of comorbidities on the association between severe infections and dementia. This paper contributes to the growing literature on infections as a risk factor for dementia. Below are a few comments to be addressed.

1. While the exposure period was 20 years in this study, it would be informative to know what the median or mean follow up was in this study.

2. Given dementia can take many years to develop, the authors could mention the potential for misclassification bias in the control group.

3. The authors could explain the rationale for the 1.20 RR cut off for selecting diseases for further analysis.

Reviewer #3:

1. Abstract: The claim of "no loss to follow-up" is overstated. Patients may not re-enter the healthcare system or may receive diagnoses outside the captured registry system, leading to potential loss to follow-up.

2. The introduction lacks a clear statement of the study's specific aims and testable hypotheses. The research objectives and expected associations should be explicitly articulated.

3. The authors describe this as a "prospective observational study," but the design is retrospective since it examines historical data from 2016-2020 to identify dementia cases diagnosed between 2017-2020.

4. The statement that controls "were also eligible to be included as cases if they had a diagnosis of dementia after the index date" raises concerns about potential misclassification bias due to delayed dementia diagnosis. How do the authors address controls who may have developed undiagnosed dementia during the study period? Especially those were choose as control cases but soon diagnosed with dementia.

5. The choice of a "two-decade exposure window 1-21 years" (line 110) lacks justification. Why was this specific timeframe selected, and how does it relate to the natural history of dementia development?

6. While the authors control for demographic variables (sex, education, marital status, employment, residence), they omit potentially important confounders such as baseline cognitive scores, medication use, and socioeconomic status. This is particularly concerning given that the identified diseases may be associated with economic factors.

7. The authors treat "area of residence, split into 18 regions" as fixed effects. Have they considered modeling regional variation as random effects, which might be more appropriate for geographic clustering?

8. The choice of "at least 1.20 for dementia" as a clinically meaningful threshold (line 165) requires brief justification or reference to established criteria.

9. The statement that "age and sex distributions were balanced between cases and controls" due to matching (lines 162-163) suggests these variables were not included in regression models. This is problematic since age and sex are potential confounders that should be controlled even in matched analyses.

10. The decision to examine only "the more commonly occurring temporal order" for disease pairs (lines 170-171) seems unnecessarily restrictive given the large sample size. Both temporal orders could be analyzed to provide more comprehensive insights.

11. The results in Figure 7 may be subject to selection bias. The authors should consider propensity score matching (PSM) or inverse probability weighting (IPW) to address potential confounding. The absence of economic status and other important covariates may bias the observed associations with infectious diseases.

12. The entire section on "disease trajectories" and disease pairs is difficult to read and would benefit from clearer description is methods and presentation in results.

---

* Please upload any figures associated with your paper as individual TIF or EPS files with 300dpi resolution at resubmission; please read our figure guidelines for more information on our requirements: http://journals.plos.org/plosmedicine/s/figures. While revising your submission, we strongly recommend that you use PLOS's NAAS tool (https://ngplosjournals.pagemajik.ai/artanalysis) to test your figure files. NAAS can convert your figure files to the TIFF file type and meet basic requirements (such as print size, resolution), or provide you with a report on issues that do not meet our requirements and that NAAS cannot fix.

After uploading your figures to PLOS's NAAS tool - https://ngplosjournals.pagemajik.ai/artanalysis, NAAS will process the files provided and display the results in the "Uploaded Files" section of the page as the processing is complete.

If the uploaded figures meet our requirements (or NAAS is able to fix the files to meet our requirements), the figure will be marked as "fixed" above. If NAAS is unable to fix the files, a red "failed" label will appear above.

When NAAS has confirmed that the figure files meet our requirements, please download the file via the download option, and include these NAAS processed figure files when submitting your revised manuscript.

* An email is required for the contact persons (who may not be co-authors) for requests to access the data.

* Please ensure that the study is reported according to the appropriate guideline and include the completed checklist as Supporting Information. When completing the checklist, please use section and paragraph numbers, rather than page numbers. Please add the following statement, or similar, to the Methods: "This study is reported as per [XXXX] guideline (S1 Checklist)."

FIGURES AND TABLES

SUPPLEMENTARY MATERIAL

REFERENCES

STUDY TYPE-SPECIFIC REQUESTS

OBSERVATIONAL STUDIES

* Abstract: Please include the study design, population and setting, number of participants, years during which the study took place (enrollment and follow up), length of follow up, and main outcome measures.

* Please ensure that the study is reported according to the STROBE (or appropriate STOBE extension) guideline (available from: https://www.equator-network.org/reporting-guidelines/strobe) and include the completed STROBE (or STROBE extension) checklist as Supporting Information. Please add the following statement, or similar, to the Methods: "This study is reported as per the Strengthening the Reporting of Observational Studies in Epidemiology (STROBE) guideline (S1 Checklist)." When completing the checklist, please use section and paragraph numbers, rather than page numbers.

* [FOR POPULATION HEALTH/REGISTRY STUDIES] Please ensure that the study is reported according to the RECORD guideline (available from https://www.record-statement.org) and include the completed checklist as Supporting Information. Please add the following statement, or similar, to the Methods: "This study is reported as per the Reporting of Studies Conducted using Observational Routinely-Collected Data (RECORD) guideline (S1 Checklist)." When completing the checklist, please use section and paragraph numbers, rather than page numbers.

* [FOR POPULATION HEALTH ESTIMATES] Please ensure that the study is reported according to the GATHER statement (available from https://www.equator-network.org/reporting-guidelines/gather-statement) and include the completed checklist as Supporting Information. Please add the following statement, or similar, to the Methods: "This study is reported as per the Guidelines for Accurate and Transparent Health Estimates Reporting (GATHER) statement (S1 Checklist)." When completing the checklist, please use section and paragraph numbers, rather than page numbers.

* [FOR MEDIATION ANALYSES] We recommend that the study is reported according to the AGReMA statement (https://agrema-statement.org/#:~:text=AGReMA%20is%20an%20evidence%2D%20and,randomised%20trials%20and%20observational%20studies) and include the completed checklist as Supporting Information. Please add the following statement, or similar, to the Methods: "This study is reported as per the Guideline for Reporting Mediation Analyses (AGReMA) statement (S1 Checklist)." When completing the checklist, please use section and paragraph numbers, rather than page numbers.

* For all observational studies, in the manuscript text, please indicate: (1) the specific hypotheses you intended to test, (2) the analytical methods by which you planned to test them, (3) the analyses you actually performed, and (4) when reported analyses differ from those that were planned, transparent explanations for differences that affect the reliability of the study's results. If a reported analysis was performed based on an interesting but unanticipated pattern in the data, please be clear that the analysis was data driven.

* Please state in the Methods section whether the study had a prospective protocol or analysis plan. If a prospective analysis plan (from your funding proposal, IRB or other ethics committee submission, study protocol, or other planning document written before analyzing the data) was used in designing the study, please include the relevant document(s) with your revised manuscript as a Supporting Information file to be published alongside your study and cite it in the Methods section. A legend for this file should be included at the end of your manuscript. If no such document exists, please make sure that the Methods section transparently describes when analyses were planned, and when/why any data-driven changes to analyses took place. Changes in the analysis, including those made in response to peer review comments, should be identified as such in the Methods section of the paper, with rationale.

MODELLING STUDIES

The following list is derived from Geoffrey P Garnett, Simon Cousens, Timothy B Hallett, Richard Steketee, Neff Walker. Mathematical models in the evaluation of health programmes. (2011) Lancet DOI:10.1016/S0140-6736(10)61505-X:

* If pertinent, please provide a diagram that shows the model structure, including how the natural history of the disease is represented, the process and determinants of disease acquisition, and how the putative intervention could affect the system.

* Please provide a complete list of model parameters, including clear and precise descriptions of the meaning of each parameter, together with the values or ranges for each, with justification or the primary source cited and important caveats about the use of these values noted.

* Please provide a clear statement about how the model was fitted to the data, including goodness-of-fit measure, the numerical algorithm used, which parameter varied, constraints imposed on parameter values, and starting conditions.

* For uncertainty analyses, please state the sources of uncertainties quantified and not quantified [can include parameter, data, and model structure].

* Please provide sensitivity analyses to identify which parameter values are most important in the model. Uncertainty estimates seek to derive a range of credible results on the basis of an exploration of the range of reasonable parameter values. The choice of method should be presented and justified.

* Please discuss the scientific rationale for the choice of model structure and identify points where this choice could influence conclusions drawn. Please also describe the strength of the scientific basis underlying the key model assumptions.

* For studies that develop a prediction model or evaluate its performance, please ensure that the study is reported according to the TRIPOD statement (https://www.equator-network.org/reporting-guidelines/tripod-statement) and include the completed checklist as Supporting Information. Please add the following statement, or similar, to the Methods: "This study is reported as per the Transparent Reporting of a Multivariable Prediction Model for Individual Prognosis Or Diagnosis (TRIPOD) statement (S1 Checklist)." For studies using machine learning, please use the TRIPOD-AI checklist. When completing the checklist, please use section and paragraph numbers, rather than page numbers.

---

## [Decision Letter · Decision Letter 2]

15 Dec 2025

Dear Dr Sipilä,

Many thanks for submitting a revision of your manuscript "Severe infections and risk of dementia in Finland: A nationwide study of the role of non-infectious comorbidities" (PMEDICINE-D-25-02079R2) to PLOS Medicine. The paper has been seen again by two of the reviewers; in addition, we recruited an additional reviewer; their comments are included below and can also be accessed here: [LINK]

As you will see, the two original reviewers state you satisfactorily addressed their concerns, but the new reviewer 4 has some additional comments. After discussing the paper with the editorial team and an academic editor with relevant expertise, I'm pleased to invite you to revise the paper in response to the reviewers' comments. We plan to send the revised paper to some or all of the original reviewers, and we cannot provide any guarantees at this stage regarding publication.

We ask that you submit your revision by Jan 05 2026 11:59PM. However, if this deadline is not feasible, please contact me by email, and we can discuss a suitable alternative.

Don't hesitate to contact me directly with any questions (sbruijn@plos.org).

Best regards,

Suzanne

Suzanne De Bruijn, PhD

Associate Editor

PLOS Medicine

sbruijn@plos.org

Comments from the reviewers:

Reviewer #1: We thank the authors for addressing our previous comments, especially pertaining to temporal order between dementia-related diseases.

Reviewer #4: Summary: This nationwide study used Finnish health registry data with a case control approach to identify hospital-treated conditions associated with dementia in dementia cases and matched controls aged 65 years and over. Conditions coded with two infection-associated ICD-10 codes - cystitis and bacterial infection of unspecified site - as well as 27 non-infectious conditions were robustly associated with dementia after control for sociodemographic confounders. The two infectious diseases remained associated with RRs ~1.19 after adjustment for the 27 comorbidities. Associations were stronger for early-onset dementia and seen across a broader range of infections.

Originality and importance: There is increasing evidence that severe infections at a range of sites are associated with increased dementia risk. With the long trajectory of dementia development, it is uncertain whether these associations are causal and when in the course of dementia pathogenesis they act. This is an innovative and well-designed study that adds to the evidence that severe infections are likely to be independent dementia risk factors. It suggests that they act relatively late, potentially accelerating the onset of clinical dementia among those with pre-clinical or possibly prodromal dementia, rather than initiating neuropathological change. The study highlights the importance of prevention and rapid treatment of severe infections in older people to prevent cognitive and functional decline.

Specific comments:

- The aim stated at the end of the introduction was to assess the extent to which the association between severe infections and dementia risk remains present after accounting for preceding non-infectious comorbidities. It was not clear how the second part of the data analysis i.e. constructing disease trajectories across dementia-related diseases fit within that aim, especially when all non-infectious dementia-related diseases were adjusted for in the final analysis

- For the disease trajectory analysis, step 2 was to identify up to 5 matched controls per disease case. In the description of this process on p13, it was unclear whether every possible combination of disease pair was assessed i.e. was each ICD-10 code taken in turn as the disease? How successful was matching i.e. was it always possible to identify up to 5 individuals without the disease? Was the most common temporal order determined by the data or by any clinical input and a priori hypotheses?

- Clarify the meaning of 'directed edges' on p13 as it is not a term well-understood by clinicians

- It was unclear how ICD-10 codes that dealt with similar/ the same conditions were considered. If investigated separately as if they were independent, would this lead to any limitations e.g. loss of power?

- In the limitations, you could note the lack of information on dementia subtypes. Previous studies have found stronger associations between severe infections and vascular dementia rather than Alzheimer's disease. It is possible that using an all-cause dementia outcome definition might have obscured some of these nuances

- The discussion did not consider why the early-onset dementia analysis identified additional infections compared to the main analysis

- On the study design diagram, figure 1, it would be helpful to show the covariate assessment window

---

* Please confirm that your title complies with PLOS Medicine's style. Your title must be nondeclarative and not a question. It should begin with main concept if possible. "Effect of" should be used only if causality can be inferred, i.e., for an RCT. Please place the study design ("A randomized controlled trial," "A retrospective study," "A modelling study," etc.) in the subtitle (ie, after a colon).

* Please confirm that your abstract complies with our requirements, including format (three sections: Background, Methods and Findings, and Conclusions) and providing all the information relevant to this study type https://journals.plos.org/plosmedicine/s/submission-guidelines#loc-abstract

* Please ensure that all abbreviations are defined at first use throughout the text.

* Please confirm that all numbers presented in the abstract are present and identical to numbers presented in the main manuscript text.

ABSTRACT

* In the abstract, please include the important dependent variables that are adjusted for in the analyses.

AUTHOR SUMMARY

* In the author summary, in the final bullet point of 'What Do These Findings Mean?', please include the main limitations of the study in non-technical language.

FIGURES AND TABLES

* When a p value is given, please specify the statistical test used to determine it in the legend.

* Please remove the P-values for the groups at baseline in Table 1.

---

## [Decision Letter · Decision Letter 3]

11 Feb 2026

Dear Dr. Sipilä,

Thank you very much for re-submitting your manuscript "Severe infections and risk of dementia in Finland: A nationwide study of the role of non-infectious comorbidities" (PMEDICINE-D-25-02079R3) for review by PLOS Medicine.

I have discussed the paper with my colleagues and the academic editor and it was also seen again by one of the reviewers. I am pleased to say that provided the remaining editorial and production issues are dealt with we are planning to accept the paper for publication in the journal.

[LINK]

We look forward to receiving the revised manuscript by Feb 18 2026 11:59PM.

Sincerely,

Suzanne De Bruijn, PhD

Associate Editor

PLOS Medicine

plosmedicine.org

Requests from Editors:

* Please change your title to: “The role of non-infectious comorbidities in the association between severe infections and risk of dementia in Finland: A nationwide registry study”

* Please remove the text ‘legend for Figure X’ from your legends.

* Please provide an explanatory sentence as legend for figure 1 and figure 2. For figure 2, ensure that you define all the abbreviations in the legend.

* Acknowledgement: as you only state that the funding information is entered in the system, please consider removing this section from the manuscript.

* Can you please clarify which commission is meant in the financial statement in the sentence: "The study does not necessarily reflect the Commission’s views and in no way anticipates the Commission’s future policy in this area.”

Comments from Reviewers:

Reviewer #4: I am satisfied that my comments have been appropriately addressed in the revised version of this manuscript.

[LINK]

---

## [Editor Report · Decision Letter 4]

13 Feb 2026

Dear Dr Sipilä,

On behalf of my colleagues and the Academic Editor, Carol Brayne, I am pleased to inform you that we have agreed to publish your manuscript "The role of non-infectious comorbidities in the association between severe infections and risk of dementia in Finland: A nationwide registry study" (PMEDICINE-D-25-02079R4) in PLOS Medicine.

PRESS

Sincerely,

Suzanne De Bruijn, PhD

Associate Editor

PLOS Medicine